# Continuous Process for Carbon Dioxide Capture Using Lysine and Tetrabutyl Phosphonium Lysinate Aqueous Mixtures in a Packed Tower

**Antonio de Jesús Zúñiga-Mendiola [1], Diana Rosa Gómora-Herrera [2],**
**Juan Manuel García-González [3] and Javier Guzmán-Pantoja [1,\*]**

[1]  Hydrocarbon Refining Management, Mexican Petroleum Institute, Eje Central Lázaro Cárdenas 152, 07730 México City, Mexico; azunigam@imp.mx

[2]  Environmental Analysis Laboratory, Mexican Petroleum Institute, Eje Central Lázaro Cárdenas 152, 07730 México City, Mexico; dianagomh@hotmail.com

[3]  Academic Unit of Chemical Sciences, Autonomous University of Zacatecas, Carretera a Guadalajara Km. 6, Ejido La Escondida, 98160 Zacatecas, Mexico; jmgarcia@uaz.edu.mx

\*  Correspondence: jguzmanp@imp.mx

**Abstract:** The $CO_2$ absorption process using aqueous solutions of lysine (Lys), the ionic liquid (IL) tetrabutyl phosphonium lysinate ([TBP][Lys]) and their mixtures was studied by means of a packed tower. The performance of these systems was evaluated through the volumetric overall mass transfer coefficient ($K_G a_V$), conducting experiments under diverse conditions such as inlet $CO_2$ concentration from 10 to 40 vol.%, gas and absorbent flow rates from 100 to 200 mL/min and from 3 to 5 mL/min, respectively, absorbent concentration from 5 to 15 wt.% and temperature from 15 to 40 °C. The obtained results for all the previous experimental conditions were better for the IL/Lys mixture than for the isolated components; the best performance was shown by the experiment varying the absorbent concentration, where the increasing $K_G a_V$ was benefited by the IL/Lys synergistic effect.

**Keywords:** ionic liquid; tetrabutyl phosphonium lysinate; lysine; $CO_2$ capture; continuous packed tower

## 1. Introduction

In recent years, the increase in carbon dioxide ($CO_2$) emissions has gained widespread attention, for it is a major contributor to global warming [1]. The major $CO_2$ sources, which represent 80% of the global emissions, are associated with electricity and heat generation, industrial activities, transport and building sectors and others [2]. According to the aforementioned, researchers have devoted their efforts not just to $CO_2$ capture, but also to its direct use or transformation in order to avoid the emission of this gas into the atmosphere [3]. As an example of the direct use of $CO_2$, the enhanced oil recovery (EOR) process has been developed [4], and the manufacture of products with high industrial demand and added value such as dimethyl ether via $CO_2$ hydrogenation [5] illustrates the case of $CO_2$ catalytic transformation [6]. These two general lines profiting from $CO_2$ capture enable the existence of circular economies by immobilizing and reducing the global concentration of this greenhouse gas. However, before making any decision regarding the two possible pathways mentioned above, the development of an efficient capture process is compelling. On the other hand, the industrial implementation of carbon capture utilization and storage (CCUS) processes is the best way to contribute to the abatement of anthropogenic $CO_2$ emissions. In this sense, China is the main producer in the world, and the concomitant CCUS technologies are still at early development stages [7]. Some proposals have been developed to couple the production and distribution infrastructure of liquefied natural gas (LNG) with

CCUS-EOR technology in order to be implemented in onshore oil fields, which would cut infrastructure costs. This $CO_2$ utilization option is the most realistic one aimed at employing CCUS in China. Another highly effective method is the geologic sequestration of $CO_2$; however, it does not represent high direct economic benefits.

The available technologies for $CO_2$ separation from gas streams are based on different processes, including adsorption, absorption, biological fixation, membrane separation, hydrate-based separation, cryogenic methods and so on. An interesting proposal was developed by Giuliano et al. [8] for the production of pure hydrogen coupled with the generation of electricity, which consisted of creating a hybrid process through a water gas shift reactor and hydrogen-selective-palladium membranes using the Selexol® process.

The most commonly used recovery methods are related to absorption–desorption processes by aqueous absorbents such as ethanolamines, ammonia and sodium or potassium hydroxide [9]. In all cases, these processes were developed and implemented on a commercial scale, and their application depends on specific conditions. In general terms, $CO_2$-bearing gases (natural gas or post-combustion gases) are passed using a countercurrent configuration through a dissolution that removes acid gases such as $CO_2$, $H_2S$ and mercaptans and retains them until desorption (regeneration processes) takes place by heating in a process unit.

In the oil industry, the most used $CO_2$ absorption process is known as Girbotol and was developed by the Girdler Corporation, USA. The medium can consist of a mono-, di- or tri-ethanolamine aqueous solution and the absorption process takes place at low temperatures (<45 °C). The $CO_2$-rich amine solution passes from the tower bottom to a heat exchanger, where the stripping process begins by passing the solution through a countercurrent to a $CO_2$/steam stream, which helps strip $CO_2$ from the aqueous amine solution; this thermal process takes place at 140 °C and this value is reached with superheated steam or with in situ reboiler equipment. In order to reuse the amine solution, it is necessary to cool down the solution in a heat exchanger to 25–40 °C. An important modification of this process occurs when the flue gases contain oxygen, which has to be removed to prevent the formation of thermally stable salts. In some cases, compounds such as piperazine could be added to increase the capacity and kinetics of the $CO_2$ absorption rate [10]. However, the main energy limitation is the high temperature needed to decompose carbamates during the regeneration step (120–140 °C).

The $CO_2$ absorption by alkanolamines remains one of the most widespread techniques in the industry for flue gases, because it is a robust and mature technology. There are several disadvantages of these alkanolamine treatments, including losses by evaporation and amine degradation due to high temperatures and energy consumption during regeneration, which form corrosive byproducts and foam, among others [11].

To overcome these problems, ionic liquids (ILs) are regarded as potential solvents for $CO_2$ absorption due to their negligible vapor pressure, thermal stability and tunable physicochemical character [11]. It is well known that imidazolium ILs are $CO_2$-philic liquids with selectivity toward other inert gases (e.g., $N_2$ and $CH_4$), and their $CO_2$ solubility increases with increasing pressure and decreases with increasing temperature. However, these ILs only enable the physical absorption of $CO_2$, requiring very high pressure (90 bar or even higher) and very long times for equilibrium to be reached (up to 24 h) [12]. The introduction of special groups, for example amino groups, can greatly enhance the $CO_2$ capture either by the anion or cation of ILs [13]. There are mainly two types of amino-functionalized ILs for $CO_2$ absorption, namely ILs where the cations are functionalized with an appended amine group and ILs in which the anions are functionalized with an appended amine group [14–17]. In our previous work [18], a study on $CO_2$ batch sorption using amino-acid-based ILs was conducted by varying cations such as tetra methyl ammonium (TMA) and tetra butyl phosphonium (TBP) and anions like glycine, alanine, valine, glutamine, histidine, arginine and lysine; the results show that the highest sorption capacity was achieved with TBP and lysine.

Based on the previous results, in the present work, the performance of $CO_2$ absorption using aqueous solutions of tetrabutyl phosphonium lysinate ([TBP][Lys]) and L-lysine (Lys) was

experimentally determined in an absorption packed tower using Pro-Pak[®] stainless steel material [19], and its operation was at counter-current and atmospheric pressure. The experiments were performed under different operating conditions in terms of absorbent concentration, liquid temperature, inert gas flow rate, inlet $CO_2$ gas concentration and liquid flow rate. The selection of mild conditions to carry out the carbamate formation reaction using specific task ILs entails various economic advantages in the implementation of this methodology at industrial level such as zero compression of the gases at the chimney exit, no conditioning by oxygen content in the gases to be treated and low regeneration temperature of the ILs.

## 2. Results and Discussion

### 2.1. IL Chemical Characterization by NMR and FTIR

The [1]H NMR spectrum is shown in Figure 1. For the phosphonium ion (cation), a triplet at 0.93 ppm, corresponding to the terminal methyl of n-butyls (A), a multiplet at 1.51 ppm, matching methyls (B and C), and a multiplet at 2.17 ppm, related to methyl next to phosphorus (D) could be observed. For lysinate (anion), a multiplet at 1.32 ppm, corresponding to methylene (1), a triplet at 2.61 ppm, corresponding to methylene (4) adjacent to the amino terminal group, and a triplet at 3.19 ppm, associated with methylene adjacent to the carboxyl group (5), were identified. The signals correlated with methylene (2 and 3) are overlapping with the signals (B and C) of the cation part at 1.52 ppm (see Figure 1).

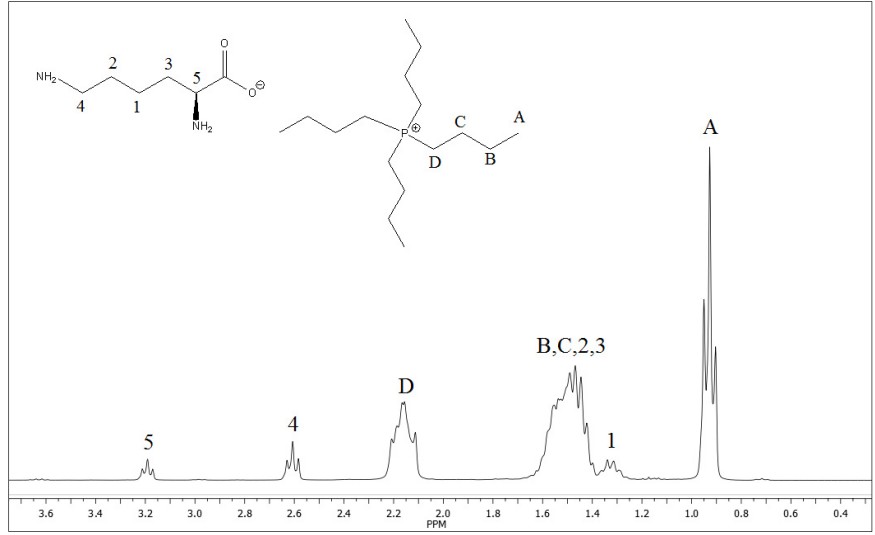

**Figure 1.** [1]H NMR spectrum of tetrabutylphosphonium lysinate. Signal **A** corresponds to hydrogens atoms from methyl group of n-butyl, signals **B** and **C** corresponds to multiplet from intermediate methyl groups and **D** corresponds to hydrogen atoms from methyl group next to phosphorus atom.

On the other hand, it is known that in aqueous solutions, the hydrogen atom bound to nitrogen (labile protons) is exchanged with hydrogen of water or, in this case, with deuterons of $D_2O$. In this sense, deuterons are not observed in [1]H NMR spectra and the signal corresponding to the amino group is not sufficiently well defined [20].

The [13]C NMR spectrum is shown in Figure 2. The butyl groups of the phosphonium ion were identified with the characteristic signals at 15.19 (A); 19.79 and 20.43 (D); 25.18 and 25.24 (B); and 25.7 and 25.9 ppm (C). Carbons B, C and D presented a double signal in the spectrum caused by the spin–spin coupling of carbon to phosphorus [21]. In addition, the characteristic signals of lysinate were observed at 24.88, 34.01, 36.92, 42.86, 58.39 and 185.79 ppm, corresponding to carbons 1–6, respectively.

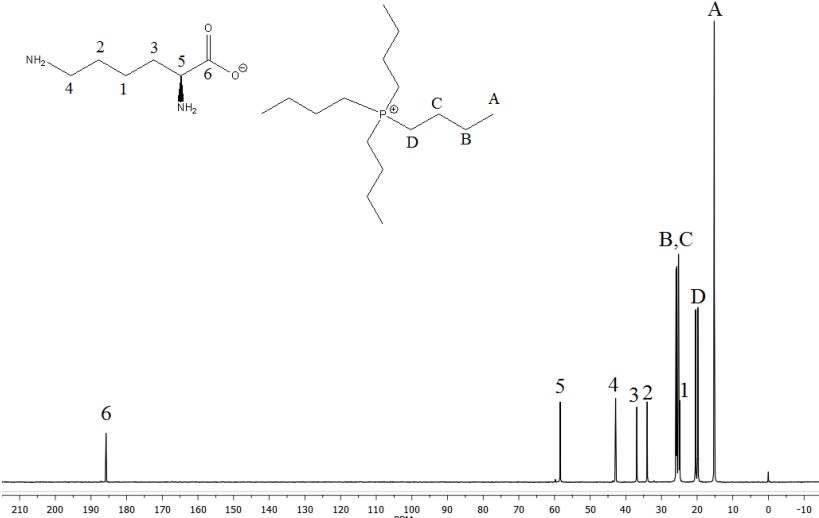

**Figure 2.** $^{13}$C NMR spectrum of tetrabutylphosphonium lysinate. Signal **A** corresponds to carbon atom in final position of butyl group. Signal **B**, **C** and **D** corresponds to carbons with double signal C-P.

In order to corroborate the structure of the synthesized IL, an FTIR study was made. In Figure 3, a spectrum of the obtained product is shown. As it can be seen, there was one peak at approximately 1574 cm$^{-1}$, which corresponded to the $CO_2^-$ group in the amino acid anion structure. In addition, the N–H stretching vibration overlapped with the OH stretching at 3357 and 3290 cm$^{-1}$ (primary amines produce two N–H peaks). The vibration at 1232 cm$^{-1}$ corresponded to the C–N stretching. The peaks at 2958, 2931 and 2871 cm$^{-1}$ were assigned to the C–H stretching vibration; bending C–H vibrations could also be observed at 1464 and 1383 cm$^{-1}$. After the NMR and FTIR analyses, it was possible to conclude that the molecular characterization of the obtained product was consistent with the [TBP][Lys] structure and in good agreement with similar results presented in the literature [22].

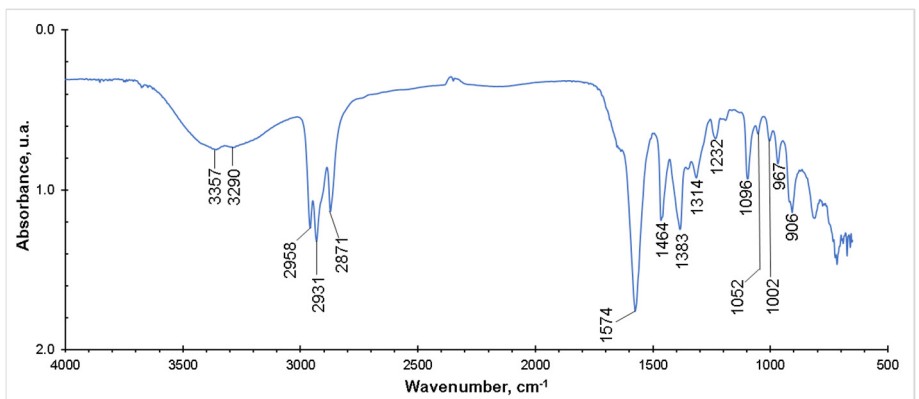

**Figure 3.** FTIR spectrum of tetrabutylphosphonium lysinate.

## 2.2. Packed-Column Performance as a Function of Operation Parameters

The overall mass transfer coefficient of the gas phase depends on the $CO_2$ inlet concentration, total gas flow rate, total liquid flow rate, aqueous IL concentration and absorbent temperature. All these effects are discussed in this section.

Regarding the pressure drop, it is known that it is an important parameter to be considered in the design of packed columns operating under vacuum or low pressure conditions and/or to establish the comparative efficiency of the geometrical design of the different packing. Notwithstanding, in the present study, its measurement was not considered because it has been reported [23] that just a slight pressure drop increase occurs with the rising viscosity of the liquid absorbent (monoethanolamine

at 30 wt.%) in a packed-bed column with a height of 5 m and 0.5 m in diameter. On the other hand, Zivdar et al. [24] reported pressure drop values from 15 to 83 mm of water per meter of gauze-type-structured packing in an air–water column. In addition, Afkhamipour and Mofarahi [25] carried out a deep review of the operation conditions of different low-pressure absorption columns and did not consider the pressure drop as a paramount operation parameter.

### 2.2.1. Effect of the $CO_2$ Inlet Concentration

As it can be observed in Figure 4, $K_G a_V$ was inversely proportional to the inlet $CO_2$ concentration at low values and had minor variations from 15 to 40 vol.% (v/v) for the IL and lysine. On the other hand, the IL + Lys mixture displayed similar behavior, but displaced to inlet $CO_2$ higher values. This behavior pattern makes sense if analyzed from a point of view where at low $CO_2$ concentrations at the column inlet the absorption efficiency was higher, and then the $CO_2$ concentration in the outlet gas was practically zero. The results obtained in the present research work are in good agreement with those reported in the literature [26,27], where it was shown that $K_G a_v$ was diminished by increasing the $CO_2$ concentration at the tower inlet (or the $CO_2$ partial pressure), which can be explained by the partial pressure gradient increase triggered by the consumption of free absorbent molecules, which in turn resulted in the coefficient decrease. The IL + Lys mixture presented higher absorption values from initial $CO_2$ concentrations of 15 vol.%.

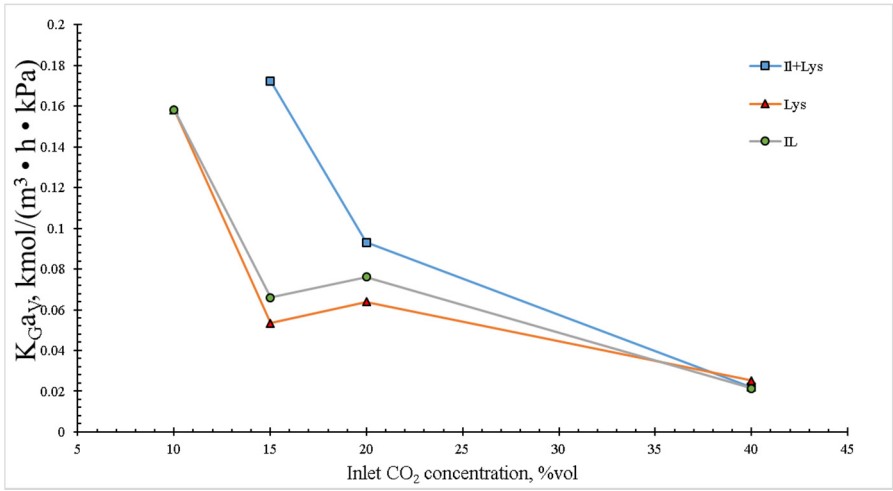

**Figure 4.** Effect of inlet $CO_2$ concentration on the overall mass transfer coefficient in the packed column. The experimental conditions were: liquid temperature of 30 °C, absorbent concentration of 10 wt.% and the total gas and liquid flow rates of 100 and 4 mL/min, respectively.

### 2.2.2. Effect of the Total Gas Flow Rate

Figure 5 shows the gas flow effect on $K_G a_V$ The maximal coefficient values were found at 100 mL/min for the three absorbent systems. By increasing the gas flow up to 200 mL/min, keeping its concentration, a decrease in $K_G a_v$ was observed. The mass transfer coefficient diminished along with the total gas flow, which corresponds to the reduction of the contact time between the gas and liquid phases at high flow rates reported in other works [26]. Another approach is related to the identification of two absorption regions [27]. In the first region, a stable coefficient with lower sensitivity to gas flow rate changes was observed. In the second absorption region, $K_G a_V$ diminished drastically with the increasing gas flow because at high rates, the absorbent was dragged, thus diminishing the liquid flow extended on the packing materials. In the case of this work, it is possible that the limits of both regions were reached, for a considerable diminution in the absorption coefficients by increasing the fed gas flow was observed. Although an increase in $K_G a_v$, promoted by higher current turbulence inside the column triggered by augmenting flow rates, was theoretically expected, which could improve

the $CO_2$ absorption, it was not observed with any of the absorbents studied in the present work. This result could be explained by the fact that the reaction kinetics of the lysine system is low and that it is a process thoroughly controlled by the mass transfer in the liquid phase, which is itself ruled by properties such as density, viscosity and chemical affinity, among others.

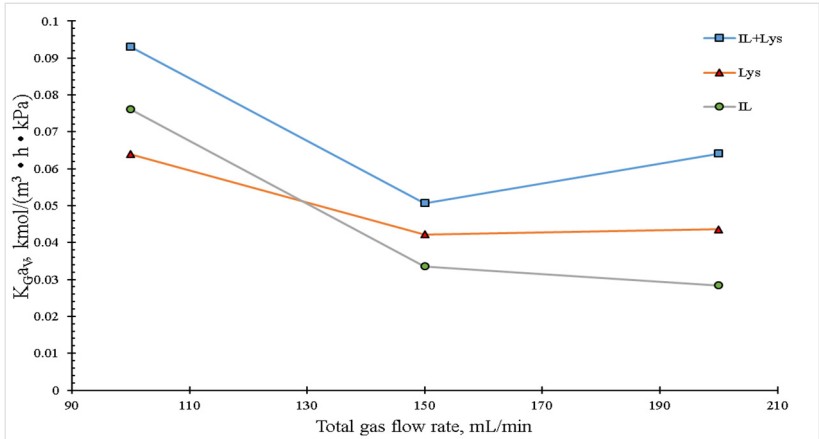

**Figure 5.** Effect of total gas flow rate on the overall mass transfer coefficient in the packed column. The experimental conditions were: liquid temperature of 30 °C, absorbent concentration of 10 wt.%, inlet $CO_2$ concentration of 20 vol.% and a liquid flow rate of 4 mL/min.

As for the global mass transfer coefficient, Fu and Zhang [28] studied an aqueous mixture of [Bmim][Gly]:MDEA with a weight ratio of 0.1:0.3 and total $CO_2$ flow of 15 vol.% using a tray tower and found that its values varied increasingly between 0.017 and 0.028 kmol/m$^3$ h kPa. In another study, Fu et al. [29] obtained values close to 0.3 kmol/m$^3$ h kPa using MEA at 2 kmol/m$^3$, Dixon rings with random configuration and $CO_2$ concentration of 14 vol.%. Another study [30] on $K_Ga_V$ for a random packing system featuring Pall rings and MEA at 1 kmol/m$^3$ reported a value of 0.36 kmol/m$^3$ h kPa, which diminished as the gas volumetric flow was augmented for a $CO_2$ concentration of 14 vol.%. In the case of the system featured in the present work, with absorbent and $CO_2$ concentrations of 10 vol.% and 20 vol.%, respectively, the global mass transfer coefficient reached values of around 0.10 kmol/m$^3$ h kPa, which means that the absorbent system proposed in this study had better performance for $CO_2$ removal. This result can be explained by the fact that lysine has two amine groups and glycine only one; these groups are mainly responsible for the sorption of acid gases in the IL structure. By analyzing the data reported in the literature when using MEA as absorbent in random packing towers, it was found that the values of the mass transfer coefficient were higher than those shown in this work and in the case of the tray tower, the reported values were lower than those featured here. From the study of the experimental parameters here described, it can be stated that their values could be further improved if an optimization process of the experimental conditions were carried out.

### 2.2.3. Effect of the Liquid Flow Rate

Figure 6 shows the effect of the liquid flow rate on the overall mass transfer coefficient. The dependence between these two variables is clear, for an increase in the liquid flow rate resulted in an increase in the $K_Ga_V$. Similar trends have been reported in the literature for packed columns using aqueous ammonia [9] or absorption towers [31–34]. If the liquid flow rate is increased, more liquid can be spread in the interfacial packing area, resulting in an increase in the overall mass transfer coefficient.

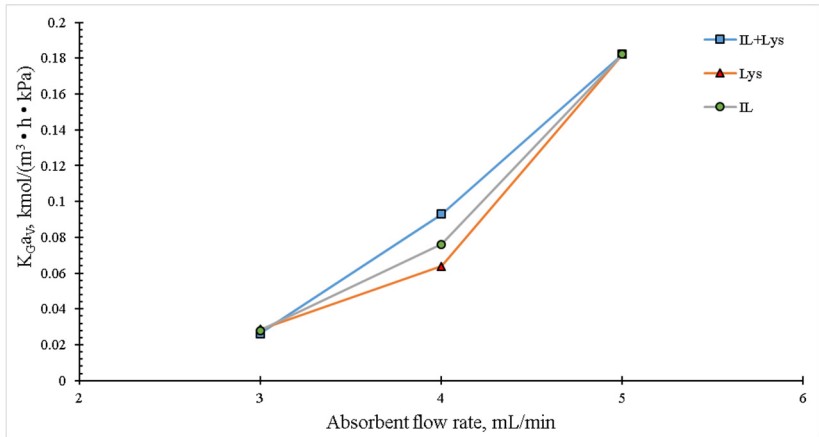

**Figure 6.** Effect of the liquid flow rate on the overall mass transfer coefficient in the packed column. The experimental conditions were: liquid temperature of 30 °C, absorbent concentration of 10 wt.%, inlet $CO_2$ concentration of 20 vol.% and gas flow rate of 100 mL/min.

### 2.2.4. Effect of the Absorbent Concentration

Figure 7 shows the mass concentration effect of the liquid phase on the overall mass transfer coefficient. An increase in the solvent concentration results on a higher $K_G a_V$, and this tendency can be observed in the different studied absorbents. This is attributed to the fact that the increasing concentration of the liquid phase increases the possibility of $CO_2$ reaction, since there are more available absorbent molecules for reaction. On the other hand, it is well known that the increase in the concentration of a liquid absorbent also augments its viscosity. However, due to the nature and low concentration of the absorbents used in the present work, no decrease in $K_G a_V$ within the studied interval was observed. In all the cases of the aqueous systems proposed in this work, their low concentrations resulted in low viscosity and then in negligible mass transfer resistance.

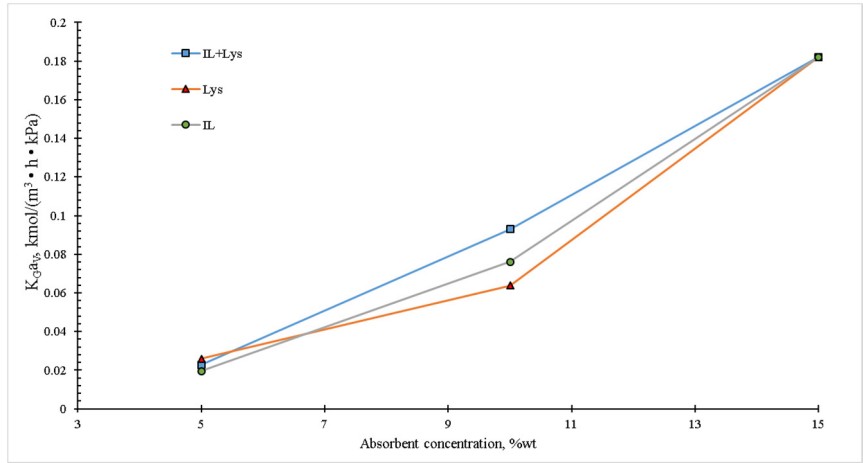

**Figure 7.** Effect of the absorbent concentration on the overall mass transfer coefficient in the packed column. The experimental conditions were: liquid temperature of 30 °C, inlet $CO_2$ concentration of 20 vol.% and total gas and liquid flow rates of 100 and 4 mL/min, respectively.

Figure 7 shows the behavior of the global mass transfer coefficient, where a monotonic increasing trend with a maximum value of 0.1873 kmol/m$^3$ h kPa occurring at an absorbent concentration of 15 wt.% could be observed. This performance was better than the one reported by Zhang et al. [34], since these authors found values between 0.0195 and 0.025 kmol/m$^3$ h kPa using a concentration of 50 wt.% with various combined absorbents (MDEA + AAIL).

### 2.2.5. Temperature Effect

The temperature at which the gas absorption takes place is one of the most important parameters for the column design, for it affects the reaction kinetics. Figure 8 shows the temperature effect on $K_G a_V$. At low temperatures (15 °C), the highest values of the global mass transfer coefficient for each absorbent system can be observed. By increasing the temperature, an evident decrease in the coefficient is obtained, which would suggest that the reaction kinetics for this lysine-based system is affected by temperatures above the room one. The same trend is displayed by the amino acid precursor, IL and their mixtures.

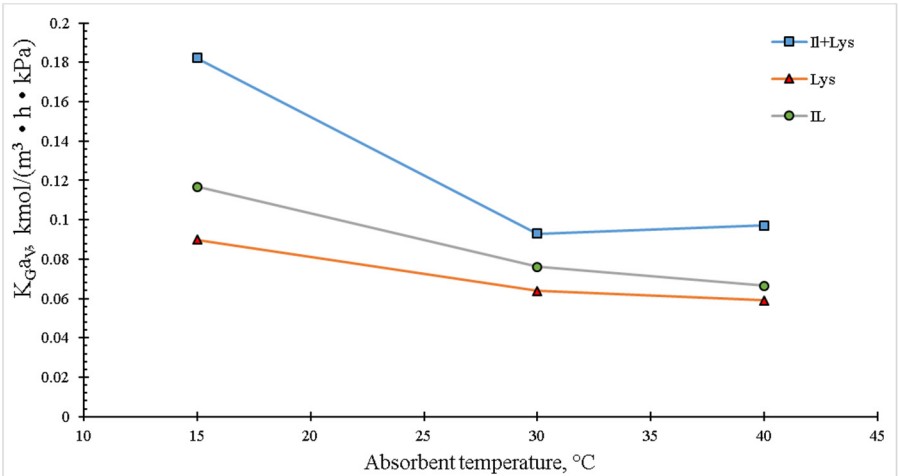

**Figure 8.** Effect of the absorbent temperature on the overall mass transfer coefficient in the packed column. The experimental conditions were: inlet $CO_2$ concentration of 20 vol.%, liquid concentration of 10 wt.% and total gas and liquid flow rates of 100 and 4 mL/min, respectively.

From the analysis of the results shown in Figures 4–8, where the behavior of the $K_G a_V$ is presented as a function of the operational parameters of the absorption column, it was observed that its value was higher for the IL + Lys case. This synergistic effect was due to the high affinity between the amino groups present in lysine and $CO_2$ to form the corresponding carbamate. It is important to note that each lysine and/or lysinate molecule has two amino groups available for the absorption of $CO_2$. In this sense, Privalova et al. [35] reported similar tendencies for the $CO_2$ sorption using three different systems: 15 wt.% MDEA + 5 wt.% PZ, 15 wt.% MEA and 1-butyl-3-methylimidazolium acetate; in all the cases, the sorption performances diminished with increasing temperatures because the dissolution reactions were inhibited, since the equilibrium constants tended to decrease with a temperature rise.

### 2.3. Performance of Absorption/Regeneration Cycles

An aspect of great importance in absorption processes is that their efficiency does not decrease significantly with the number of use cycles. In the present work, the methodology for carrying out absorption and regeneration cycles in the packed tower was implemented under the following conditions: the absorption took place at 30 °C with total gas and absorbent flow rates of 100 and 4 mL/min, respectively, $CO_2$ inlet concentration of 15 vol.% and [TBP][Lys] concentration of 10 wt.%. Regeneration was carried out at 95 °C with total gas (nitrogen) and spent absorbent flow rates of 40 and 4 mL/min, respectively, and [TBP][Lys] concentration of 10 wt.%. It is important to note that the optimum regeneration temperature was 100 °C. However, this was not possible in the continuous unit due to the employed liquid flow rates and the fact that the column reboiler was slightly pressurized, which produced a liquid phase when entering the tower at atmospheric pressure; for this reason, the selected experimental temperature was 95 °C.

The absorbent behavior during five sequential regeneration cycles is shown in Figure 9. It can be seen that in the first cycle, the $CO_2$ removal efficiency in the input current was 80%; at the second cycle,

it was 76% and in the following cycles, it stabilized at 75%. Therefore, the implemented regeneration and reuse conditions were suitable for the efficient operation of the absorption column.

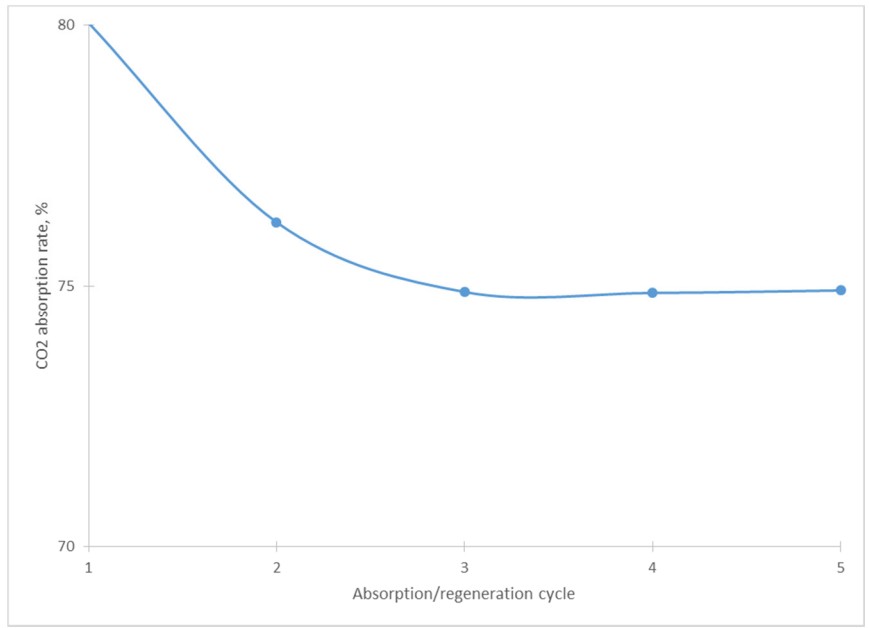

**Figure 9.** Absorption/regeneration behavior for aqueous absorbent [TBP][Lys] at 10 wt%.

## 3. Experimental Section

### 3.1. Materials

High purity $CO_2$ (≥99.99%) and $N_2$ (≥99.99%) were purchased from Infra Group (México City, Mex.). Tetrabutyl phosphonium bromide (≥98%) and acetonitrile (anhydrous, 99.8%) were acquired from Sigma-Aldrich (Darmstadt, Germany). All other chemicals and solvents were reagent grade and used as received without further purification. Deionized water (0.05 µS/cm) was used throughout the experiments.

### 3.2. Ionic Liquid Synthesis

[TBP][Lys] was prepared by adapting the methodology described by Pernak et al. [17]. The main modification consisted in diminishing the temperature to protect the hydroxide intermediate since it tends to decompose to yield a tertiary phosphine oxide and alkane under alkaline conditions (β-elimination). The formed precipitate is insoluble in ethanol and was separated by filtration. One molar equivalent of potassium hydroxide was added to 1 molar equivalent solution of tetrabutyl phosphonium bromide in ethanol and stirred for 2 h in a controlled temperature bath (−5 °C). The next step was represented by the addition of 1.1 molar equivalent of L-lysine (10 wt.% excess) to the previous solution, stirring for 5 h at low temperature (−5 °C). Afterwards, the solvents were distilled at 60 °C under vacuum, and the unreacted reagents were precipitated with acetonitrile. The solution was filtered and the solvent was evaporated at 80 °C under vacuum. Finally, the product was dried at 80 °C under reduced pressure for 2 days.

### 3.3. Analysis Method

The chemical structure of the IL was determined by $^1H$ and $^{13}C$ nuclear magnetic resonance (NMR) with deuterated water as solvent. A JEOL Eclipse-300 (Tokyo, Japan) equipment was used for $^1H$ (300 MHz) and $^{13}C$ (75.5 MHz) NMR. Additionally, its structure was corroborated by FTIR spectroscopy using a Nicolet 8700 (Wilmington, DE, USA) piece of equipment with DTGS detector

from Thermo Scientific. Before the analysis, the IL was dried overnight under vacuum conditions. The analysis was performed at room temperature and 50 scans with a resolution of 4 cm$^{-1}$.

The $CO_2$ concentration at the outlet of the packed column was analyzed by means of an FTIR spectrometer (Nicolet PROTÉGÉ 460; Wilmington, DE, USA) that uses a DTGS detector with KBr windows. This spectrometer has a vacuum cell for gases with 10 m of optical path length. The cell was maintained at a constant temperature of 50 °C. The absorption was followed by sampling every 8 min, enough time to ensure the cell purging between samples, until the equilibrium was reached. The obtained spectra were analyzed in absorbance in the interval ranging from 4000 to 400 cm$^{-1}$ and the area under the curve was correlated with the $CO_2$ concentration.

### 3.4. Experimental Apparatus and Procedures

#### 3.4.1. Packed-Column Absorption Process

The scheme of the $CO_2$ absorption system is shown in Figure 10. The packed column (3) had a height of 30 cm and 2.5 cm of inner diameter and was filled with stainless steel Pro-Pak® [19] (State College, PA, USA) featuring 0.41 mm (0.16 inch) in diameter with the same length value. The characteristics of this material are displayed in Table 1. The packed section had a height of 15 cm and the liquid inlet had 2 cm above the packing materials. The absorption system was operated at counter-current and atmospheric pressure. A correct absorbent distribution is of great importance to achieve high absorption efficiencies; in this study the aqueous solutions were injected using 5 hypodermic needles mounted at the top end of the column.

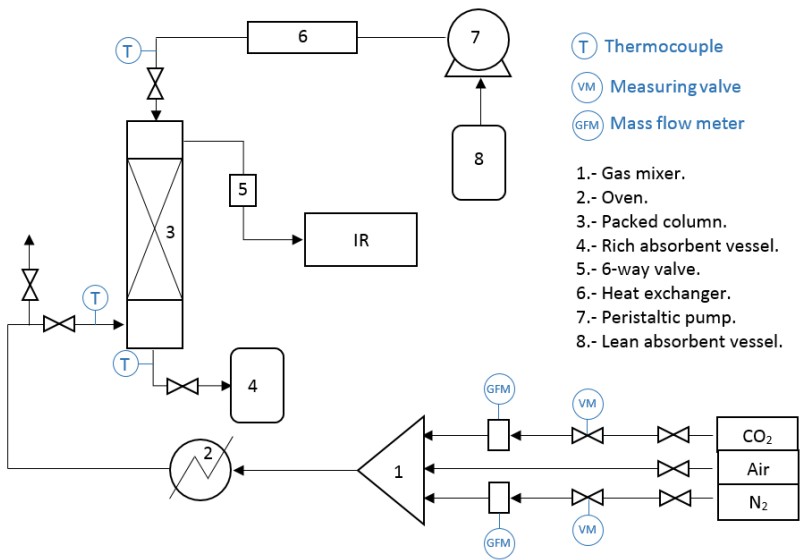

**Figure 10.** Continuous $CO_2$ absorption system.

**Table 1.** Characteristics of the Pro-Pak® packing material [19].

| Size, mm | Material | Free Space,% | Surface Area, m$^2$/m$^3$ | Packing Factor |
|:---:|:---:|:---:|:---:|:---:|
| 0.41 | Stainless steel | 94 | 1890 | 693 |

The gas flow rate was measured by a mass flow meter (Aalborg, GFM-17; Orangeburg, NY, USA). The gas currents were mixed in a stainless steel sphere (1), 10 cm in diameter, filled with Raschig rings with 0.63 cm in diameter. If needed, the gas could be pre-heated in an oven (2) (CI model ALF, México City, Mex.), which featured a 0.63 cm tubing spiral with 4.5 m in length, filled with stainless steel grills to maximize gas heating. All the tubing connections were isolated using mineral wool (thickness of 2.5

cm). The gas mixture temperature was measured with thermocouples type J (Watlow; Chicago, IL, USA). Afterwards, the gas mixture entered the packed column from the bottom.

The absorbent liquid was circulated with a peristaltic pump (7) (Barnant, type HD MA; Barrington IL, USA) with a maximum capacity of 1 L/h. Prior to the admission into the packed column, the liquid was heated or cooled, as the case may be, in a concentric tube heat exchanger (6) constructed with 1/2 (outer tube) and 1/4 inch (inner tube) stainless steel tubes. The heat exchanger was operated at counter-flow. Afterwards, the absorbent liquid entered the packed column from the top.

For the experiments, the lean absorbent was pumped into the column to reach the desired temperature and operation flow rate. To avoid channeling the gas to the liquid outlet, the column bottom was filled with liquid until it reached 2 cm of height. When the operating conditions were reached, the gaseous mixture was circulated through the packed column. The absorbent loaded with $CO_2$ left the column at the bottom and was directed to a vessel to be stored. The operation conditions are presented in Table 2.

**Table 2.** Operation conditions to evaluate the $CO_2$ absorption by IL, Lys and their mixtures.

| Parameter | Operation Conditions |
|---|---|
| Absorbent | [TBP][Lys], L-lysine, mixtures |
| Gas flow rate | 100, 150, 200 mL/min |
| Liquid flow rate | 3, 4, 5 mL/min |
| $CO_2$ concentration | 10, 15, 20, 40 vol.% |
| Absorbent concentration | 5, 10, 15 wt.% |
| Liquid temperature | 15, 30, 40 °C |

### 3.4.2. Batch-Cell-Absorbent-Regeneration Experiments

The experiments to determine the appropriate temperature for the regeneration of the $CO_2$ absorbent system were performed in a batch cell described in our previous publication [18], which consists of measuring the pressure changes as a function of time for a fixed volume of absorbent at controlled temperature. For a given temperature (80, 90, 100 and 120 °C), 3 absorption and regeneration cycles were carried out. In general, the absorption test ends when the $CO_2$ partial pressure remains constant, and in the case of desorption, a heating process at atmospheric pressure with vapor recovery is implemented for 30 min at each tested temperature. These experiments were aimed at identifying the regeneration temperature to be implemented during the packed-column tests. The experimental results are shown in the Supplementary Materials Section. Figure S2 shows the results of the average of three measurements for absorbent regeneration cycles [TBP] [Lys] at 10 wt.% as a function of temperature from 80 to 120 ° C.

### 4. Determination of the Overall Mass Transfer Coefficient in Packed Column

The volumetric overall mass transfer coefficient, $K_G a_V$, is a significant parameter to evaluate the absorption process of gases in semi- or continuous equipment from the laboratory scale to industrial level. The analysis of the mass transfer phenomenon is based on the two-film theory and under steady-state conditions, the absorbed mass flux of component $A$ ($N_A$) across the gas–liquid interface can be represented in terms of $K_G$ [31] as follows:

$$N_A = K_G\left(P_{y_A} - P_{y_A^i}\right) \tag{1}$$

where $K_G$ is the overall mass transfer coefficient for the gas phase; $P_{y_A}$ is the partial pressure at the gas bulk; and $P_{y_A^i}$ is the partial pressure at the gas–liquid interface. Therefore, it is more practical to determine the absorption performance based on the unit volume of the packed column rather than on the interface area as follows:

$$N_A a_V = K_G a_V P\left(y_A - y_A^*\right) \tag{2}$$

where $y_A^*$ is the partial pressure of component A in equilibrium with the concentration of A in the liquid bulk. Considering a packed column element with height $dZ$ (m), the mass balance equation, according to the rate-based model, can be expressed as follows:

$$N_A a_V dZ A_c = Gd\left(\frac{y_A}{1 - y_A}\right) \tag{3}$$

where $G$ is the inert gas flow rate (kmol/h); $A_C$ is the column cross-section area ($m^2$); P is the total system pressure (kPa). By substituting Equation (3) into Equation (2), $K_G a_V$ (Kmol/$m^3$ h kPa) can be calculated as follows:

$$K_G a_V = \left(\frac{G}{P\left(y_A - y_A^*\right)A_C}\right)\frac{dY_A}{dZ} \tag{4}$$

The values of $K_G a_V$ can be obtained by measuring the inlet and outlet concentrations (mol fraction) of the gas phase as [25]:

$$K_G a_V = \left(\frac{\frac{G}{A_C}}{P\left(\frac{y_{A,in} - y_{A,out}}{\ln\frac{y_{A,in}}{y_{A,out}}}\right)}\right)\left(\frac{Y_{A,in} - Y_{A,out}}{Z}\right) \tag{5}$$

## 5. Conclusions

The molecular structure of [TBP][Lys] was corroborated by means of NMR and FTIR studies. The continuous $CO_2$ absorption behavior in a packed column was carried out through the calculation of the global mass transfer coefficient using a specific task ionic liquid (IL) and its components in aqueous solution. This IL was dissolved in water, and its $CO_2$ absorption behavior revealed that its mass transfer efficiency is proportional to its concentration. It can also be seen that the overall mass transfer coefficient in the three liquid absorbent systems is higher in the case of the [TBP] [Lys] mixed with lysine for all the studied parameters.

The most suitable temperature to carry out the regeneration of the spent IL was 95 °C. This value is lower than the one reported for other traditional $CO_2$ absorption systems based on alkanolamines; then, its potential use could represent a competitive advantage, which based on the results presented in this study, encourages the recommendation of employing this IL in packed columns at an industrial level to carry out the removal of $CO_2$.

**Supplementary Materials:** The following are available online at http://www.mdpi.com/2073-4344/10/4/426/s1, Figure S2. $CO_2$ reabsorption rate as a temperature function using [TBP][Lys] at 10 wt.%. The data shown is the average of three regeneration cycles.

**Author Contributions:** A.d.J.Z.-M. carried out the experimental work, contributed to the writing, answered questions from the Reviewers and improved the manuscript. D.R.G.-H. performed the fine tuning of analytical techniques. J.M.G.-G. supported with resources obtained for APC and with the global structure of the investigation. J.G.-P. designed the research and was responsible for writing and coordinating all the tasks related to the paper publication. All authors have read and agreed to the published version of the manuscript.

**Funding:** This research was funded by Mexican Petroleum Institute grant number D.61072 and the APC was funded by The Directorate for Academic Improvement (Dirección de Superación Académica, DGESI-SEP, México) under the responsibility of Lic. Lorenzo Manuel Loera de la Rosa.

**Acknowledgments:** This work was supported by the Mexican Petroleum Institute through the D.61072 Project ascribed to the Hydrocarbon Refining Management. A.J. Zúñiga-Mendiola thanks CONACyT for the granted Ph.D. scholarship.

**Conflicts of Interest:** The authors declare no conflict of interest.

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
