# Peer review of "Continuous Process for Carbon Dioxide Capture Using Lysine and Tetrabutyl Phosphonium Lysinate Aqueous Mixtures in a Packed Tower"

_catalysts, doi:10.3390/catal10040426_

Round 1

Reviewer 1 Report

The manuscript by and Antonio and coworkers reported the optimal design of lysine and tetrabutyl phosphonium lysinate aqueous mixtures for carbon dioxide separation. Process in a packed tower system was employed in carbon dioxide absorption. Overall mass transfer coefficient was evaluated as a function of operation parameters, including CO2 inlet concentration, total gas flow rate, liquid flow rate, absorbent concentration and temperature. However, the study in this manuscript doesn’t match with study scope of “Catalysts” Journal. The review would like to recommend manuscript author to re-submit manuscript to Journal of “Process” or “Applied Sciences”.

Author Response

We do thank the Reviewer’s observation and suggestion. As this Catalysts special issue is focused on the presentation of results stemming from CO2 areas such as capture, transformation, uses and/or reduction of related gas emissions, we consider that our work is suitable for this special issue, since it deals with a CO2 capture process using lysine and tetrabutylphosphonium lysinate aqueous mixtures.

Reviewer 2 Report

In this work, the authors assessed the performances of the ionic liquid tetrabutyl phosphonium lysinate for CO2 absorption. The study was carried out in a lab-scale apparatus and the volumetric overall mass transfer coefficient was estimated, as a function of several parameters, such as gas/liquid volumetric flow, CO2 concentration and temperature. The paper is well-written but major revision is recommended. Below, the main comments are reported.

  • The authors should emphasize the catalytic conversion of the recovered CO2, in order to better fit with the Journal’s topics. The following papers concerning CCSU should be cited:Quadrelli et al., Carbon Dioxide Utilis. (2015) 285;Giuliano et al., Int. J. Hydrogen Energy 43 (2018) 19279;Catizzone et al., Molecules, 23 (2018) 31;Centi and Perathoner, ChemSusChem 7 (2014) 1274.
  • Line 112: please check if “10 m” is correct;
  • Line 136: the pump maximum capacity is 1 L/min, while the tests are carried out up to 5 mL/min. Did the authors check the accuracy of the pump for these low values of flowrate?
  • Line 173: the equation (4) is valid for constant value flowrates. In the study, G value changes because the relatively high CO2 concentration adopted. Please, recalculate the overall mass transfer coefficients by properly integrating the equation (3).
  • Figure 5 and 6: the authors should give a convincing explanation of the obtained results.
  • Line 238 on: Zhang et al. studied the absorption in tray tower system. The overall mass transfer should be compared for similar towers. The following studies may be used as reference:Naami et al., Int. J. Greenhouse Gas Contro, 19 (2013) 3-12;Dey et al. Energy Procedia 1 (2009) 211-215;Tan et al., J. Ind. Eng. Chem. 18 (2012) 1874-1883.
  • Line 262-269: the authors should revise this part. The increase of absorbent concentration improves the overall mass transfer coefficient, despite the viscosity increase (that is not measured).

Author Response

The authors should emphasize the catalytic conversion of the recovered CO2, in order to better fit with the Journal’s topics. The following papers concerning CCSU should be cited: Quadrelli et al., Carbon Dioxide Utilis. (2015) 285; Giuliano et al., Int. J. Hydrogen Energy 43 (2018) 19279; Catizzone et al., Molecules, 23 (2018) 31; Centi and Perathoner, ChemSusChem 7 (2014) 1274.

Author’s Response (AR):

Thank you very much for your observation and in order to comply with it, the following paragraph was added to the Introduction: “According to the aforementioned, researchers have devoted their efforts not just to CO2 capture, but also to the direct use or transformation in order to avoid the emission of this gas into the atmosphere [3].  As an example of the direct use of CO2, the enhanced oil recovery (EOR) process is found [4] and the manufacture of products with high industrial demand and added value illustrates the case of CO2 catalytic transformation [5].  These two general lines profiting from the CO2 capture enable the existence of circular economies by immobilizing and reducing the global concentration of this greenhouse gas. However, before making any decision regarding the two possible pathways mentioned above, the development of an efficient capture process is compelling.”

Line 112: please check if “10 m” is correct;

AR: This value is correct. The optical path length for the employed cell is 10 m.

Line 136: the pump maximum capacity is 1 L/min, while the tests are carried out up to 5 mL/min. Did the authors check the accuracy of the pump for these low values of flowrate?

AR: The Reviewer is right. The correct value is 1 L/h, which means 16 mL/min. The corrected value has been featured in the manuscript.

Line 173: the equation (4) is valid for constant value flowrates. In the study, G value changes because the relatively high CO2 concentration adopted. Please, recalculate the overall mass transfer coefficients by properly integrating the equation (3).

AR: We agree with the Reviewer. Eq. 4 is valid only for cases where the gas flow values are constant. In our calculations, we confirmed that the volumetric flow of nitrogen did not change during the absorption process due to the inert nature of this gas, so the reported values of the global mass transfer coefficient are correct. This methodology is reported in the literature to calculate this coefficient for both packed [Energy Procedia 1 (2009) 211-215] and spray towers [Energy Proc. 2017, 114, 1665-1670].

Figure 5 and 6: the authors should give a convincing explanation of the obtained results.

AR: We agree with the Reviewer. To comply with the observation, the following paragraph was added to offer a better description of the information depicted in Figures 4 and 5 (the Figure’s number changed after the modification of the Section Results):

Information on Figure 4 (Section 2.2.1): “The results obtained in the present research work are in good agreement with those reported in the literature [20, 21], where it is shown that KGav diminished by increasing the CO2 concentration at the tower inlet (or the CO2 partial pressure), which can be explained by the partial pressure gradient increase triggered by the consumption of free absorbent molecules, which in turn results in the coefficient decrease.”

Information on Figure 5 (Section 2.2.2): “The mass transfer coefficient diminished along with the total gas flow, which corresponds to the reduction of the contact time between the gas and liquid phases at high flow rates reported in other works [20]. Another approach is related to the identification of two absorption regions [22]. In the first region, a stable coefficient with lower sensitivity to gas flow rate changes is observed.  In the second absorption region, KGaV diminished drastically with the increasing gas flow because at high rates, the absorbent is dragged, thus diminishing the liquid flow extended on the seals. In the case of this work, it is possible that the limits of both regions be implied, for a considerable diminution in the absorption coefficients by increasing the fed gas flow is observed.”

Line 238 on: Zhang et al. studied the absorption in tray tower system. The overall mass transfer should be compared for similar towers. The following studies may be used as reference: Naami et al., Int. J. Greenhouse Gas Contro, 19 (2013) 3-12; Dey et al. Energy Procedia 1 (2009) 211-215; Tan et al., J. Ind. Eng. Chem. 18 (2012) 1874-1883.

AR: We do thank the Reviewer’s observation and suggestion. The following lines were added as follows:

Section 2.2.2, page 8, line 261: “In another study, Fu et al. [24] obtained values close to 0.3 Kmol/m3 h kPa using MEA at 2 Kmol/m3, Dixon rings with random configuration and CO2 concentration of 14 vol.%. Another study [22] on KGaV for a random seal system featuring Pall rings and MEA at 1 Kmol/m3 reported a value of 0.36 Kmol/m3 h kPa, which diminished as the gas volumetric flow augmented for a CO2 concentration of 14 vol.%.”

Section 22.2, page 8, line 272: “By analyzing the data reported in the literature when using MEA as absorbent in random seal towers, it was found that the values of the mass transfer coefficient are higher than those shown in this work and in the case of the tray tower, the reported values are lower than those featured here.”

Line 262-269: the authors should revise this part. The increase of absorbent concentration improves the overall mass transfer coefficient, despite the viscosity increase (that is not measured).

AR: We thank the Reviewer’s observation. In order to clarify this point, the following lines were added as follows:

Section 2.2.4: “It is well known that the increase in the concentration of a liquid absorbent also augments its viscosity. However, due to the nature and low concentration of the absorbents used in the present work, no decrease in KGaV within the studied interval was observed.”

PS: some of Section, page, line, and figure numbers were changed after the revision process.

Reviewer 3 Report

 This work is well performed, clearly presented and novel Minor revision is suggested so that the authors also correlate their results with literature, e.g. Privalova et al. https://doi.org/10.1039/C2RA23013E and other.

Author Response

We do thank and agree with the Reviewer´s suggestion. Our results were correlated with those obtained by Privalova et al. as follows:

Section 2.2.5: In this sense, Privalova et al. [30] reported similar tendencies for the CO2 sorption using three different systems: 15 wt. % MDEA + 5 wt. % PZ, 15 wt. % MEA and 1-butyl-3-methylimidazolium acetate; in all the cases, the sorption performances diminished with increasing temperatures because the dissolution reactions were inhibited, since the equilibrium constants tend to decrease with a temperature rise.”

Round 2

Reviewer 1 Report

The reviewer would like to appreciate the point-to-point response from the author. But the reviewer does not understand why this manuscript was re-sent to Catalysts. This manuscript reports about absorption of carbon dioxide with ionic liquid. The reviewer insists that the scope of study falls out of “Catalysts”. The reviewer would like to recommend the manuscript to be published on Processes or Applied Science.

Reviewer 2 Report

The paper has been only partially improved. The present version of manuscript is not suitable for publication due to some points which have to be addressed:

1) In first revision, the reviewer suggested some papers useful for the introduction part. At least the suggested papers should be reported. In addition to these, the authors may add some other papers concerning CCUS. This is important for fitting the topic of the Journal. 

2) The authors should indicate or estimate the pressure drop inside the tower in order to properly integrate equation (4). The authors should  discuss such aspect.

3) Equations: the units of all the parameters should be added.

Author Response

1) In first revision, the reviewer suggested some papers useful for the introduction part. At least the suggested papers should be reported. In addition to these, the authors may add some other papers concerning CCUS. This is important for fitting the topic of the Journal.

Author’s Response (AR):

In the previous version, 2 out of 4 of the references suggested by the Reviewer were included and another one related to CCUS was added. In this new version, the two missing suggested references were featured and another one dealing with CCUS, published in 2018 by Adu et al., was also considered. The Introduction was enriched as follows:

Page 1, line 40:

“On the other hand, the industrial implementation of carbon capture, utilization and storage (CCUS) processes is the best way to contribute to the abatement of anthropogenic CO2 emissions. In this sense, China is the main producer in the world and the concomitant CCUS technologies are still at early development stages [7]. Some proposals have been developed to couple the production and distribution infrastructure of liquefied natural gas (LNG) with the CCUS-EOR technology in order to be implemented in onshore oil fields, which would cut infrastructure costs. This CO2 utilization option is the most realistic one aimed at employing CCUS in China. Another highly effective method is the geologic sequestration of CO2; however, it does not represent high direct economic benefits.”

Page 2, line 51:

“An interesting proposal was developed by Giuliano et al. [8] for the production of pure hydrogen coupled with the generation of electricity, which consisted in creating a hybrid process through a water gas shift reactor and hydrogen-selective-palladium membranes using the Selexol® process.”

2) The authors should indicate or estimate the pressure drop inside the tower in order to properly integrate equation (4). The authors should discuss such aspect.

AR: The authors thank the insightful previous and present Reviewer’s comments. In this research work, due to experimental conditions such as low gas and absorbent concentrations and low height of the packed bed (0.15 m), it can be considered that the measurement of the pressure drop would be highly uncertain according to literature reports [23-25]. Due to the importance of this design parameter, the following paragraph was added to Section 2.2:

Page 4, line 144:

“Regarding the pressure drop, it is known that it is an important parameter to be considered in the design of packed columns operating under vacuum or low pressure conditions and/or to establish the comparative efficiency of the geometrical design of the different packing. Notwithstanding, in the present study, its measurement was not considered because it has been reported [23] that just a slight pressure drop increase occurs with the rising viscosity of the liquid absorbent (monoethanolamine at 30 wt.%) in a packed-bed column with a height of 5 m and 0.5 m in diameter. On the other hand, Zivdar et al., 2006 [24] reported pressure drop values from 15 to 83 mm of water per meter of gauze-type-structured packing in an air-water operating column. In addition, Afkhamipour and Mofarahi, 2017 [25], carried out a deep review of the operation conditions of different low-pressure absorption columns and did not consider the pressure drop as a paramount operation parameter.”

3) Equations: the units of all the parameters should be added.

AR: Thank you very much for the observation. The units of all the parameters were added to the text. Were included in page 12, lines 371-375.

PS: the changes made from revision #1 are marked in yellow and from revision #2 are in green.

Round 3

Reviewer 2 Report

The manuscript has been improved. The new version is suitable for publication.